# The Impacts of Environmental Dynamism on Chinese Tour Guides’ Sustainable Performance: Factors Related to Vitality, Positive Stress Mindset and Supportive Policy

**DOI:** 10.3390/ijerph19159289

**Published:** 2022-07-29

**Authors:** Ping Sun, Xiaoming Zhou, Cui Shao, Wenli Wang, Jinkun Sun

**Affiliations:** School of Management, Shandong University, Jinan 250100, China; spmay@sdu.edu.cn (P.S.); 202013163@mail.sdu.edu.cn (X.Z.); 202113123@mail.sdu.edu.cn (C.S.); 202113127@mail.sdu.edu.cn (W.W.)

**Keywords:** COVID-19, tour guide, environmental dynamism, sustainable performance, vitality at work, positive stress mindset, supportive policy

## Abstract

Although previous research shows great interest in improving the sustainability of organizations’ performance, little is known about individual sustainable performance, especially for special groups such as tour guides. Drawing on the Conservation of Resources (COR) theory, this study aimed to investigate the effect of environmental dynamism caused by COVID-19 on tour guides’ sustainable performance and mediating role of vitality and intervention mechanism in this relationship. Adopting a quantitative research method, we collected data from 382 professional tour guides in China via three surveys. The Structural Equation Model (SEM) and PROCESS were used to test the hypotheses. The results revealed that: (1) environmental dynamism was negatively related to tour guides’ sustainable performance and (2) vitality at work mediated this negative effect; (3) a positive stress mindset moderated the relationship between environmental dynamism and vitality; (4) supportive policy’s moderating role in the relationship of vitality and sustainable performance was not significant. The above conclusions contribute to the literature about the external environment, emotional state, performance management and application boundary of COR theory in the context of the COVID-19 pandemic.

## 1. Introduction

When first impacted by the COVID-19 pandemic, the Chinese government implemented strong pandemic prevention measures, including restrictions on travel [1]. Although the pandemic has been effectively controlled, international travel is still forbidden and even inter-provincial travel is disallowed when infections occur again [2], which keeps China’s tourism industry from fully recovering. Tourism workers are the most directly affected because of the closure of tourism. They are faced with unpaid leave or dismissal during the crisis period [3]. Being frontline employees and always playing multiple roles during the journey, tour guides are an important part of the tourism industry in China, where organized tours still enjoy great popularity [4]. Previous research on tour guides mostly paid attention to how to improve service quality [5] and enhance visitor experience [6], with a lack of attention to tour guides’ individual behavior and psychological traits in the context of COVID-19. Lots of studies focused on the impacts of the pandemic on the tourism industry and organizations [7,8] or the impacts on tourists’ tourism preferences [9] and behavior [10]. The literature about the effects of COVID-19 on psychological behavior and employee performance in the hospitality industry remains scant [11]. It is unclear what kinds of impacts have been composed on tour guides’ physical and mental well-being, as well as their sustainable performance [3].

Some previous studies have discussed that the turbulence caused by COVID-19 might lead to individual job uncertainty and insecurity [12]. It has been proved that COVID-19 may lead to psychological stress and have negative impacts on frontline workers’ job satisfaction and work performance [13,14]. According to the Conservation of Resources (COR) theory, individuals have the motivation to protect their current resources and acquire new resources, which can help us understand tour guides’ psychological changes under the impacts of COVID-19. Tour guides may need to invest their physical and psychological resources to deal with the negative impacts of the crisis [13]; therefore, it is necessary to discuss whether the decrease in resources reduces tour guides’ vitality at work. Being generally believed beneficial to physical and mental health, vitality has a positive effect on individual self-control performance and creativity [15]. Vitality is defined as “maximizing work performance as well as worker health and well-being” [16]; employee sustainable performance is often overlooked in the academic literature [17]. Thus, it is crucial to investigate vitality’s impacts on tour guides’ performance, especially sustainable performance.

Moreover, little is yet known about how to mitigate the negative impacts of COVID-19 on tour guides’ vitality and sustainable performance. Previous studies mainly focus on the support from organizations [12,18], while tour guides could obtain little support from their organizations (travel agencies) considering the practical situation. As one of the hardest hit sectors of the tourism industry, most travel agencies were closed or even bankrupted during the crisis period. In addition, the Chinese tourism administration has implemented reforms for freelance tour guides, which further affects the relationship between tour guides and travel agencies. The current study thus mainly focuses on individual and social-level factors that can mitigate the negative impacts of COVID-19. Studies have proved that an individual stress mindset has a distinct influence on the response to stress [19]. In practice, those tour guides who have a positive stress mindset could actually respond to the stress of COVID-19 more positively, such as trying to play live streaming and short-form videos and learning new skills in China. From the social perspective, the government’s supportive policy should play an important role, especially in China. Supportive policies usually have preferential features in the context of the pandemic [20], which may help tour guides maintain vitality and have a positive performance.

This study aims to explore the impacts of environmental dynamism caused by COVID-19 on tour guides’ vitality at work and sustainable performance by conducting a series of surveys about Chinese tour guides. We intended to determine the relationship between vitality and sustainable performance, as there is still no consensus about it in existing research [17,21]. More especially, this research explores the moderating roles of the positive stress mindset and supportive policy from individual and social levels, which is in line with practice and could expand the study of boundary conditions of the relationships between crisis impacts and individual performance theoretically. Based on the above analyses, the moderated mediation model is given to test the following hypotheses, as shown in Figure 1.

### 1.1. The Relationship between Environmental Dynamism and Tour Guides’ Sustainable Performance

Environmental dynamism is defined as the rate of change and the degree of instability of the environment [22,23,24]. Low environmental dynamism means that the market demand, technological changes and institutional environment are stable and predictable, while high environmental dynamism is the opposite [25]. Similar to the current pandemic environment, the external environment of enterprises is full of uncertainty, and customer demand is constantly shrinking or changing [25]. As an important external factor of enterprises, environmental dynamism’s moderating effects on organizational innovation [26], new product development [27], new venture performance [28] and other enterprise performance have been tested by many scholars, while only a few empirical articles have focused on the direct impact of environmental dynamism on enterprise performance or frontline workers’ performance based on individual perspective [29,30,31]. The environment of the tourism industry is unpredictable and fragile, especially during the COVID-19 pandemic [32]. Tour guides are always in direct contact with tourists and have the most intuitive observation of the tourism market. As a result, they could perceive the environmental dynamism caused by the pandemic more clearly—job insecurity and work pressure—which will affect their mental health and increase their anxiety and depression [32]; therefore, it is particularly necessary to consider the direct impact of environmental dynamism on tour guides’ work emotion and performance.

In recent years, more and more researchers have realized the necessity to study the performance of organizations as well as employees from a sustainable perspective [17]. The concept of employee sustainable performance (E-super) was proposed because some researchers believe that employees fall into the trap of increasing demand for work and decreasing work resources and job compensation [16,33,34,35]. For the E-super of tour guides, it is crucial to understand sustainability and their performance [36,37,38]; however, in the existing tour guide-related studies, researchers mainly focused on the impacts of tour guides’ performance on the sustainable behavior of tourists, or on the dimensions of identifying tour guide performance [39,40]. In the context of the turbulence of the industry, the study of the sustainable performance of tour guides is of great practical significance for the development of tour guides.

Hence, the following hypothesis is proposed:

**Hypothesis** **1** **(H1).***Environmental dynamism negatively affects tour guides’ sustainable performance*.

### 1.2. The Mediating Role of Vitality at Work

Vitality at work is a positive emotional state and is defined as high levels of energy, effort, persistence and resilience while working [41]. What exactly affects vitality has always been the focus of research. Shraga et al. [42] have explored vitality’s work-related antecedents, including meaningful interactions with others, specific job characteristics (e.g., job significance), feedback from supervisors and job identity. Furthermore, vitality at work is thought to be influenced by dispositional and contextual variables, and motivational process variables of initiating and sustained behavior at work [43]; however, existing research rarely explored vitality from the perspective of external environmental factors, so this study focuses on this research gap.

Vitality is one of the dimensions of work engagement, which is deemed to be a strictly positive and relatively stable indicator of work well-being [44]. Work engagement is considered to be the opposite aspect of burnout [44], and so is vitality at work. According to COR theory and previous research, high environmental dynamism leads tour guides to face the risk of resource loss, resulting in burnout [45]. Then, vitality will decline accordingly. In terms of vitality itself, it is considered an overall evaluation of job demand and available physical, emotional and cognitive resources [43], so tour guides naturally have a negatively cognitive appraisal when their resources are lost or at risk. In conclusion:

**Hypothesis** **2a** **(H2a).***Environmental dynamism negatively affects tour guides’ vitality at work*.

Having vitality means being healthy and having organic well-being [14,46]. Tremblay et al. [47] have proved that in stressful environmental disasters, individuals with high vitality experienced less physical and psychological damage. It was found that vitality could improve employability [48,49]. Vitality represents a highly supportive mechanism for career success [50]. In a work environment, vitality can advance individual work performance [51] and stimulate individual creativity by bringing inspirational thoughts and behavior [52,53]. The higher of employee’s work vitality, the fewer mistakes when performing a task [54].

As we have mentioned above, there is no consensus yet about the relationship between tour guides’ vitality and sustainable performance. Some researchers believe that vitality is an important part of sustainable performance [15,21]. Others think vitality has independent effects on sustainable performance. In the current study, we argue that vitality has a positive influence on the tour guides’ sustainable performance. Taken together, we further expect that environmental dynamism has a negative impact on tour guides’ sustainable performance by reducing their vitality at work.

Therefore, we propose the following hypotheses:

**Hypothesis** **2b** **(H2b).***Vitality at work positively affects tour guides’ sustainable performance*.

**Hypothesis** **2c** **(H2c).***Vitality at work mediates the relationship between environmental dynamism and tour guides’ sustainable performance*.

### 1.3. The Moderating Role of Positive Stress Mindset

Stress is often seen as negative and destructive, while the individual mindset towards stress can influence their response to stress [19]. A stress mindset refers to an employee’s view of how stress affects their life and how to respond to it [3]. A positive stress mindset is the extent to which employees hold the mindset that stress can be a source of personal growth, well-being and performance [55].

Based on COR theory, a positive stress mindset can be seen as an important psychological resource [3], which can help individuals deal with the stress of resource losses. A positive stress mindset’s moderating role in the context of stress has been preliminarily revealed in some special groups [56]. For example, it has been proved that a positive stress mindset can help adults reduce their perceived distress and act less impulsively when facing adversity [57]. In Tuan and Trong’s study [58], a positive stress mindset positively influenced a salesperson’s resilience to a crisis such as COVID-19. So positive stress mindset may protect tour guides from suffering stress in the context of COVID-19. Thus, we conclude that a positive stress mindset can buffer emotional depletion from environmental dynamism.

**Hypothesis** **3** **(H3).***Positive stress mindset moderates the relationship between environmental dynamism and vitality such that this relationship is weaker when tour guides have a high positive stress mindset*.

### 1.4. The Moderating Role of Supportive Policy

Policy support from government administrations and other external assistance can be decisive in restoring national livelihoods [59]. Supportive policy in this study refers to a series of helpful policies introduced by the government to deal with the impacts of COVID-19 on tour guides, which can promote the development of national livelihood recovery strategies and reduce the impacts of the pandemic on livelihood capital [20]. For example, the Chinese government has issued a series of departmental documents such as “Notice on Work Matters Related to Actively Responding to the Impact of the Pandemic and Maintaining the Stability of the Tour Guide Team”. According to COR theory, individuals with more resources are less likely to be attacked by resource losses and are more able to access resources [60]. Supportive policy from the government can be seen as a supplement to tour guides’ resources and amplify the positive impacts of their individual resources; therefore, the importance of vitality will be enhanced if they have a perception of supportive policies. On this basis, we put forward the following hypothesis:

**Hypothesis** **4** **(H4).***Supportive policy moderates the relationship between vitality and sustainable performance of tour guides such that the relationship is stronger when the supportive policy is high*.

## 2. Research Methods

### 2.1. Study Design and Participants

We designed a time-lagged study with three data collection intervals for tour guides from different provinces of China. China’s tourism industry suffered a huge loss as the first victim of COVID-19 [61]. As an important part of the tourism industry, tour guides almost lost their work at the beginning of the pandemic; therefore, we conducted our first data collection in December 2020 when COVID-19 continued to be severe. At Time 1, tour guides completed measures of environmental dynamism and individual positive stress mindset. The Chinese government implemented a series of supportive policies, and we conducted Time 2 data collection in July 2021 to measure tour guides’ vitality and supportive policy. Six months later, they completed our measurement of sustainable performance at Time 3.

We invited 659 Chinese tour guides, who had qualification certificates only, which allows them to be professional tour guides in China, to fill in our questionnaires with the help of the tourism association and the association of travel services. With limited offline activities due to the pandemic, we sent our survey to tour guides via email and WeChat (social communication software widely used in China). Participants were told that their responses would remain confidential. A total of 412 tour guides responded to all three survey waves of our survey, among which 382 were usable. Among the 382 participants, 54.7% were female, 50.8% were aged between 31–40 and 72.3% have worked as a tour guide for more than ten years. It was worth noting that while the participants were from 27 different provinces (China has a total of 34 provincial-level administrative regions), there were 140 participants from Shandong Province, which accounts for more than a third; therefore, we employed an independent-samples *t*-test using SPSS 26.0 to inspect whether the mean values of the main variables were significantly different between Shandong Province and other provinces. The *t*-test results showed that none of the means was significantly different at 95% confidence level (*p* values of five variables’ *t*-test were 0.950, 0.488, 0.781, 0.315 and 0.859), which would not interfere with further data analysis.

### 2.2. Measures

We used a five-point Likert scale (1: “strongly disagree”; 5: “strongly agree”) to measure the variables and used the back-translation method to ensure scale translation quality [62]. One researcher translated the existing scales into Chinese; then, they were back-translated into English by another member of the research team. We adapted the existing scales according to the working situation of Chinese tour guides.

Environmental dynamism. Following the measurement items adopted by Jansen and Van den Bosch [25], four items were taken to measure environmental dynamism in the background of COVID-19, such as “Environmental changes in our local tourism market are intense”; “In our local tourism market, changes are taking place continuously”. The Cronbach’s alpha of the variable scale was 0.873 in this current study.

Vitality at work. We chose the scale developed by Ronit Kark [63], in which vitality at work included 6 items such as “I am most vital when I am at work”; “I am full of positive energy when I am at work” and “When I am at work, I feel a sense of physical strength”. The Cronbach’s alpha of the variable in the current study was 0.885.

Positive stress mindset. According to the measurement scale of scholars Crum et al. [19], with the actual situation of this study combined, the measurement items of positive mindset included “The effects of this stress are positive and should be utilized”; “Experiencing this stress improves my health and vitality”; “Experiencing this stress enhances my performance and productivity” and so on. The Cronbach’s alpha of the variable was 0.898.

Supportive policy. Supportive policy was measured by three items from a scale established by Zhao [20]. A sample item includes “The government has provided us with a lot of employment information and job opportunities to deal with the pandemic”. The Cronbach’s alpha of the supportive policy in this study was 0.883.

Tour guides’ sustainable performance. We used a developed ten-item scale to measure tour guides’ sustainable performance [17]. Sample items were: “During my entire tour guide career, I will be able to continuously achieve the objectives of my job”; “During my entire tour guide career, I will be able to permanently meet the criteria for my job performance” and so on. The Cronbach’s alpha of tour guides’ sustainable performance in current study was 0.962.

Control variables. In line with In-Jo Park [64], we controlled for demographic variables including gender, age, level of certificate (measured by the level of Certificate of Tour Guide, which includes four levels in China), working years as a tour guide and so on.

## 3. Data Analysis and Results

### 3.1. Common Method Bias Test

We tried to avoid the common method bias problem in the data collection phase by separating the measurements of our main five variables (environmental dynamism and positive stress mindset were measured at Time 1; vitality at work and supportive policy were measured at Time 2; tour guides’ sustainable performance was measured at Time 3). However, the common method bias may still exist because our data were collected from the same source (measurements of all variables were completed by tour guides). Therefore, we examined common method bias using Harman’s single-factor test, with which all items of the questionnaire were analyzed by factor analysis. Consistent with previous researchers’ recommendations, if the factor analysis of all variables generated a single factor that explains more than 50% of the variance, the data might have a serious common method bias [65]. Results of factor analysis showed that the first factor explained 45.99% of the variance, which suggested that common method bias was not a significant problem in the current study.

### 3.2. Descriptive Statistics and Correlation Analysis

Table 1 presents the means, standard deviations and correlations of the current research variables. The bivariate correlations indicated that environmental dynamism was negatively correlated with tour guides’ vitality and sustainable performance (r = −0.25, *p <* 0.01; r = −0.36, *p <* 0.01, respectively). We also found that vitality was significantly correlated with sustainable performance (r = 0.57, *p <* 0.01).

### 3.3. Preliminary Analysis

We conducted a Confirmatory Factor Analysis (CFA) to assess the scale validity using AMOS 21.0. In line with the recommendations of previous researchers [64], the measurement model can be accepted if CFI ≥ 0.90, TLI ≥ 0.90, IFI ≥ 0.90, RMSEA ≤ 0.08 and SRMR ≤ 0.06. The results (see Table 2) showed that the fit of the five-factor model was acceptable (χ^2^/df = 2.27, *p* = 0.00; NFI = 0.920; CFI = 0.953; RMSEA = 0.058; SRMR = 0.035). All factor loadings were greater than 0.7 significantly, ranging from 0.71 to 0.92. The five variables’ CR values were 0.87, 0.90, 0.88, 0.89 and 0.96. The AVE values of these five variables were 0.64, 0.70, 0.72, 0.61 and 0.72. Then we compared the fitting degree of the five-factor model with the various nested models to further investigate the discriminative validity of the five latent variables. As the results showed in Table 3, our hypothesized five-factor model had the best fit compared to all the other alternative models, which further proved that the five variables had reasonable discriminative validity.

### 3.4. Hypothesis Testing

Firstly, we conducted a Structural Equation Model (SEM) to test the hypothesized direct and indirect effects using AMOS 21.0, and the path coefficients of the main variables are shown in Figure 2. The path coefficient between environmental dynamism and tour guides’ sustainable performance was significantly negative (β = −0.267, *p <* 0.001), which provided evidence for the negative effect of environmental dynamism on tour guides’ sustainable performance—H1 was supported. H2a about the negative effect of environmental dynamism on vitality at work and H2b about the positive effect of vitality at work on tour guides’ sustainable performance were significantly supported (β = −0.340, *p <* 0.001; β = 0.522, *p <* 0.001).

Using the regression-based bootstrapping approach, the SPSS PROCESS macro plugin developed by Hayes was widely used in the mediation and moderation analyses [66,67]. We thus applied SPSS PROCESS 3.3 to further test the mediating effects and moderating effects. Firstly, we employed PROCESS (5000 bootstrap resamples) to test the mediating role of vitality at work. As a resampling method, bootstrapping can efficiently reduce type 1 error methodologists [68]. Bootstrapping results (Table 4) showed that 95% Confidence Interval (CI) varied from −0.177 to −0.076 without zero being included in the interval, which demonstrated that vitality at work mediated the effect of environmental dynamism on tour guides’ sustainable performance (indirect effect = −0.124, *p <* 0.001, 95% CI = [−0.177, −0.076]). Thus, it was more convinced that vitality at work played a mediating role in the current study—H2c was supported.

Similar to the test of mediating role above, we used PROCESS to test the moderating of a positive stress mindset and supportive policy. It was suggested that the interaction effect was supported if the 95% confidence interval of interaction terms did not include zero [67]. As is shown in Table 5, the 95% confidence interval of the interactions term of environmental dynamism did not include zero (β = −0.14, *p* < 0.01, 95% CI = [−0.24, −0.04])—Hypothesis 3 was supported. In order to observe the moderating effect of positive stress mindset more intuitively, referring to previous studies, we used the point which varied above and below the mean one standard deviation of a positive stress mindset to draw the interactive effect diagram of environmental dynamism and positive stress mindset (Figure 3). As can be seen from Figure 3, the influence of environmental dynamism on the vitality with high positive stress mindset was lower than that of tour guides with a low positive stress mindset.

However, beyond our expectations, the confidence interval of the interaction of supportive policy and vitality varied from −0.06 to 0.10 with zero included, which indicated that the moderating role of supportive policy on sustainable performance was not significant (β = 0.02, *p*= 0.61, 95% CI= [−0.06, 0.10]), thus Hypothesis 4 was not supported. To sum up, tour guides’ individual positive stress mindset played a moderating role, but the moderating effects of supportive policy were not reflected in this study.

## 4. Discussion

In this study, we explored the impacts of COVID-19 on tour guides’ vitality and sustainable performance. Drawing on COR theory, we proposed our hypotheses and established a study framework based on the practical situation and theoretical reasoning. We assumed environmental dynamism caused by the COVID-19 pandemic was an antecedent of vitality and could further influence sustainable performance. At the same time, we chose individual positive stress mindset and supportive policy as moderating variables. We drew these conclusions after the empirical test of Chinese professional tour guides.

First, we proved that tour guides had an obvious high perception of environmental dynamism in the context of the pandemic crisis, which directly led to a significant decrease in tour guides’ vitality and sustainable performance. The above results were in line with our expectations. Tour guides have been severely affected by COVID-19 as frontline workers of the most hit industry, which was consistent with existing research about other types of frontline workers [69,70,71,72].

Second, we revealed that environmental dynamism could also indirectly influence tour guides’ sustainable performance via vitality’s mediating role. Our test indicates that vitality at work had a positive relationship with sustainable performance, while environmental dynamism in the context of COVID-19 made workers perceive job insecurity [67] and emotional exhaustion [73], which led to a decrease in vitality at work. Accordingly, tour guides would have a negative sustainable performance as the decrease in vitality due to the intense environmental dynamism.

Third, an individual positive stress mindset could weaken the negative impact of environmental dynamism on vitality. Specifically, with a high level of positive stress mindset, tour guides may believe that they can obtain positive consequences from experiencing stress. Thus, their vitality may not decrease so obviously compared to those who had a negative stress mindset.

Finally, not consistent with our hypothesis, the supportive policy cannot moderate the relationship between vitality and sustainable performance according to our test result, which was also not consistent with JieYin’s expectation of the role of supportive policy [67].

### 4.1. Theoretical Implications

Based on COR theory, we revealed the impacts of environmental dynamism (caused by the COVID-19 pandemic) on tour guides’ vitality and sustainable performance, which means we extended the application of COR theory in the study of crisis events. To our knowledge, there was no prior research analyzing the emotional state and sustainable performance of tour guides in the context of COVID-19. Previous research mainly focused on the impacts of the pandemic on industries or organizations such as airlines [74], hotels [75] and so on, or researchers were interested in how the organizational changes impact their workers’ behavior and performance in the context of COVID-19 [76]. So, there still remains a gap about how COVID-19 directly affects tourism frontline workers [3], and we tried our best to fill it by revealing the relationship between environmental dynamism, individual vitality and sustainable performance.

We identified the relationship between vitality and sustainable performance. Unlike most studies that paid attention to employees’ negative psychological state in the crisis [77], we noticed the role of vitality at work and explored its relationship with performance from a sustainable perspective. As we have mentioned before, some scholars hold the belief that vitality is an important aspect of employee sustainability [78]; in the current study, we proved vitality and sustainable performance are separate and vitality could affect it. This result strongly supported Tianchang Ji et al.’s point: as an expression of well-being, vitality was a crucial determinant of an employee’s sustainable performance [17]. Moreover, this study also found a new antecedent of vitality—environmental dynamism—which means we enriched the theoretical study of vitality.

We explored how to buffer the negative impact of environmental dynamism and how to improve sustainable performance from an individual and social support perspective. We identified the important role of a positive stress mindset as a vital individual psychological resource. This finding provided a new direction about how to help frontline workers experience the crisis. While supportive policy’s moderating role was not significant in this study, we still added to the literature on intervention policies, which filled the gap of existing literature rarely considering the importance of policy [67] and established a foundation for further research. We further analyzed why the supportive policy’s importance was not significant in this study. One possible reason was that Chinese governments have been implementing relatively strict restrictive policies for quite a long time while introducing supportive policies, which have limited the recovery of tourism. Thus, these restrictive policies may negatively affect tour guides’ perception of supportive policies, which leads to the statistic of supportive policy not exactly reflecting reality and affecting our results. Another possible reason was that lots of supportive policies are aimed toward industries or organizations instead of tour guides directly; therefore, the individuals had no obvious perception of supportive policies.

### 4.2. Managerial Implications

The results showed that the outbreak and persistence of COVID-19 have significant impacts on tour guides’ vitality at work and sustainable performance. There are some suggestions for tourism administrations and organizations in order to help tour guides suffering the crisis and retain the valuable human resources for the tourism industry.

Because most tour guides have a substantial loss of income, the government should provide basic security for tour guides’ livelihood and medical care with the help of providing them job opportunities and forgiving or deferring their loans. Especially, the supportive policies should be directly targeted at them individually. What is more important, governments of different levels should timely adjust the control and restriction policies as the pandemic changes. The proper recovery of tourism is critical to the economic well-being of tour guides.

In addition, more attention should be paid to this special group, especially their psychological situation. Although their work environment and living conditions are vulnerable, the complexity of tasks and the diversity of roles as a tour guide could help them to build better psychological conditions, which may relatively promote dealing with the stress [4]. Psychological resources play an important role in helping them experience the pandemic. Accordingly, administrations and associations should try to provide free psychological counsel and support, protecting tour guides from unhealthy mental states and promoting them to have a positive stress mindset.

At last, tourism administrations and organizations should manage to improve tour guides’ vitality and performance from a sustainable perspective. Whether or not the pandemic exists, the tourism industry is always changing, which means the demands on the ability of tour guides must be improved accordingly; therefore, we suggest that administrations and associations ought to organize some public training for tour guides aiming to improve their vitality and ability to continue working in the tourism industry.

### 4.3. Limitations and Future Research Directions

First, while we collected data in three waves and conducted a common method bias test before we tested our hypotheses, all of the statistics of this study were from tour guides’ self-reports, which may lead to the deviation of the analysis results. Another limitation that cannot be ignored about the data was that the interval between different data collecting times might be too long because the antecedent variables may change widely, especially during the height of the pandemic. Thus, in a future study, we can try different methods and sources to obtain data at the appropriate collecting times. Second, we only tested the moderating role of individual positive stress mindset and supportive policy, while there may be other factors that can moderate the relationship between environmental dynamism, vitality and sustainable performance, such as family support and so on. Third, a qualitative study could be used to further clarify the causes and connections of the investigated influences and their impacts in more depth, just as Lin and other scholars’ study designs [62]. Finally, this study’s participants are from different provinces of China; in a future study, we can take the influence of regional market environment characteristics into consideration and scholars can conduct cross-culture research on other countries’ tour guides in future studies.

## 5. Conclusions

This study constructs the analysis framework of environmental dynamism and tour guides’ sustainable performance during the COVID-19 pandemic by examining the mediating mechanism of vitality at work and exploring the moderating influence of a positive stress mindset and supportive policy through extending the application of COR theory in the context of a crisis event. It contributes to the literature by changing research of environmental dynamism from a macro perspective to employees’ emotional states and enriches the research scope of sustainability of individual performance, whose antecedents are overlooked by academic and empirical studies. In addition, it confirms that vitality at work is one of the determinants of employee sustainable performance and there is an intervening effect of a positive stress mindset on the relationship between environmental dynamism and vitality at work. This study provides insights into how to overcome the negative effects on tour guides, who represent frontline workers hit heavily by turbulence caused by COVID-19, and how to build a positive emotional state and then obtain sustainability of individual performance.

## Figures and Tables

**Figure 1 ijerph-19-09289-f001:**
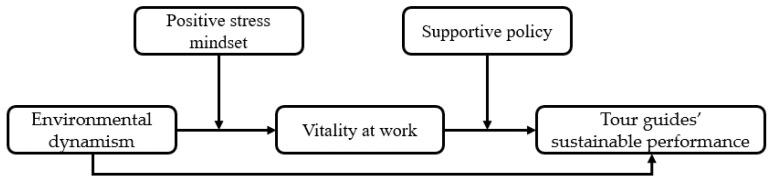
Hypothetical model of the current study.

**Figure 2 ijerph-19-09289-f002:**
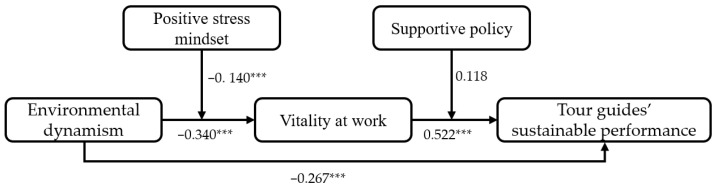
Path coefficients of the main variables. Note: *** *p <* 0.001. To make the graphics concise, Figure 2 does not present the path coefficients of control variables that were actually included in the testing model.

**Figure 3 ijerph-19-09289-f003:**
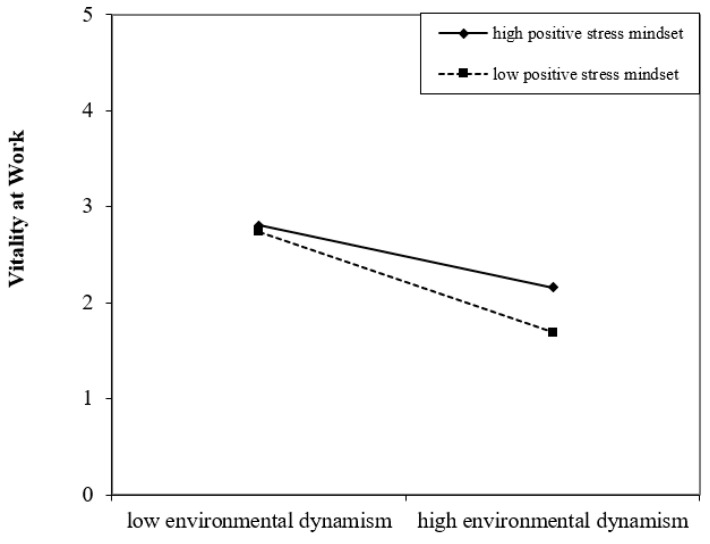
Moderating effect of positive stress mindset.

**Table 1 ijerph-19-09289-t001:** Means, standard deviations and correlations.

Variable	Mean	SD	1	2	3	4	5	6	7	8	9
1. Gender	1.55	0.50	1								
2. Age	3.33	0.83	−0.31 **	1							
3. Working years	4.29	1.39	−0.18 **	0.65 **	1						
4. Level of certificate	2.22	1.02	−0.17 *	0.46 **	0.46 **	1					
5. Environmental dynamism	4.19	0.79	−0.20	−0.53	0.53	−0.41	1				
6. Positive stress mindset	3.69	0.91	−0.02	−0. 01	0.13 *	0.01	−0.41 *	1			
7. Supportive policy	2.13	1.07	0.03	−0.18 **	−0.18 **	−0.20	−0.24 *	0.32 **	1		
8. Vitality at work	4.04	0.79	−0.47	0.26 **	0.26 **	0.12 **	−0.25 **	0.55 **	0.17 **	1	
9. Sustainable performance	4.09	0.77	0.73	0.19 **	0.19 **	0.28	−0.36 **	0.73 **	0.26 **	0.57 **	1

Note: N = 382, Two-tailed test; * *p <* 0.05, ** *p <* 0.01.

**Table 2 ijerph-19-09289-t002:** Confirmatory factor analysis results.

Variable	Mean	Factor Loading	Cronbach’s Alpha	AVE	CR
Environmental dynamism (ED)			0.87	0.64	0.87
Environmental dynamism in our local tourism market changes in tourism market are intense	4.24	0.78			
In our local tourism market, changes are taking place continuously	4.24	0.86			
In our local tourism market, the volumes of products and services to be delivered change fast and often	4.15	0.80			
Our clients regularly ask for new products and services	4.12	0.74			
Positive stress mindset (PSM)			0.90	0.70	0.90
The effects of the stress from COVID-19 are positive and should be utilized	3.75	0.79			
Experiencing the stress from COVID-19 facilitates my learning and growth	3.81	0.84			
Experiencing the stress from COVID-19 enhances my performance and productivity	3.66	0.88			
Experiencing the stress from COVID-19 improves my health and vitality	3.51	0.77			
Supportive policy (SP)			0.88	0.72	0.88
The government has provided us with a lot of employment information and job opportunities to deal with the pandemic	2.16	0.82			
The government has introduced sufficient preferential policies for loans and repayments to support our response to the pandemic	2.27	0.87			
The government has introduced various subsidy mechanisms such as transportation subsidies and subsistence allowances to help us deal with the pandemic	1.96	0.85			
Vitality at work (VW)			0.89	0.61	0.89
I am most vital when I am at work	4.26	0.77			
I am full of positive energy when I am at work	3.87	0.71			
My organization makes me feel good	4.08	0.82			
When I am at work, I feel a sense of physical strength	4.00	0.81			
When I am at work, I feel mentally strong	4.05	0.81			
Tour guides’ Sustainable performance (TSP)			0.96	0.72	0.96
I will continuously achieve the objectives of my job	3.92	0.87			
I will permanently meet the criteria for my job performance	4.00	0.85			
I will continuously demonstrate expertise in all job-related tasks	3.81	0.87			
I will persistently perform well in the overall job by carrying out tasks as expected	3.63	0.85			
I will continuously fulfill all the requirements of my job	4.06	0.85			
I will permanently be competent in all areas of my job	4.05	0.86			
I will persistently manage more responsibility than typically assigned	4.16	0.92			
I will organize and plan well to achieve the objectives of my work in a sustainable way	4.10	0.89			
I will organize and plan well to meet the deadlines of my work in a sustainable way	4.14	0.75			
I will permanently be suitable for my job	4.11	0.77			

Note: N = 382; χ^2^ = 655.189, df = 289, χ^2^/df = 2.27, *p <* 0.001; NFI = 0.920; CFI = 0.953; RMSEA = 0.058; SRMR = 0.035; AVE = Average variance extracted; CR = Composite reliability; All factor loadings are significant at *p <* 0.01.

**Table 3 ijerph-19-09289-t003:** Fitting degree of competition models.

Models	χ^2^	df	RMR	NFI	RFI	IFI	TLI	CFI	RMSEA
Five-factor model	655.189	289	0.033	0.920	0.910	0.954	0.948	0.953	0.058
Four-factor model ^a^	1239.966	293	0.063	0.849	0.832	0.880	0.866	0.880	0.092
Three-factor model ^b^	1827.322	296	0.113	0.777	0.755	0.806	0.786	0.805	0.117
Two-factor model ^c^	2361.312	298	0.124	0.712	0.686	0.739	0.714	0.738	0.135
One-factor model ^d^	2833.505	299	0.135	0.654	0.624	0.679	0.650	0.678	0.149

Note: a: environmental dynamism and positive stress mindset combined; b: environmental dynamism and positive stress mindset combined, vitality at work and supportive policy combined; c: all antecedent variables combined; d: all variables combined.

**Table 4 ijerph-19-09289-t004:** Estimates and confidence intervals for the indirect effects of vitality at work.

Direct Effect			Indirect Effect		
	B	S.E.		95% CI. Lower	Upper
ED→TSP	−0.22 ***	0.05	ED→VW→TSP	−0.18	−0.08
ED→VW	−0.25 ***	0.21			
VW→TSP	0.50 ***	0.41			

Note: *** *p <* 0.001.; ED = Environmental dynamism; VW = Vitality at work; TSP = Tour guides’ sustainable performance; Bootstrap sample size = 5000.

**Table 5 ijerph-19-09289-t005:** Estimates and confidence intervals for the moderating roles.

Model	Model 1 (Vitality at Work)			Model 2 (Sustainable Performance)		
	β	S.E.	95% CI	β	S.E.	95% CI
Constant	−0.04	0.98	[−1.97, 1.89]	2.83 ***	0.43	[1.99, 3.68]
Environmental dynamism	0.54 *	0.22	[0.11, 0.97]	−0.18 ***	0.04	[−0.26, −0.10]
Positive stress mindset	1.06 ***	0.23	[0.61, 1.51]			
Vitality at work				0.43 ***	0.08	[0.26, 0.60]
Supportive policy				0.04	0.17	[−0.30, 0.39]
Environmental dynamism × Positive stress mindset	−0.14 **	0.05	[−0.24, −0.04]			
Vitality at work × Supportive policy				0.02	0.04	[−0.06, 0.10]

Note: * *p <* 0.05.; ** *p <* 0.01., *** *p <* 0.001.; Bootstrap sample size = 5000.

## Data Availability

The calculation data used in this paper come from the World Tourism Organization, Organization for Economic Co-operation and Development, and International Energy Agency, which have been explained in the main text.

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
