# Peer review of "The Impacts of Environmental Dynamism on Chinese Tour Guides’ Sustainable Performance: Factors Related to Vitality, Positive Stress Mindset and Supportive Policy"

_ijerph, 2022, doi:10.3390/ijerph19159289_

Round 1

Reviewer 1 Report

The study is focused on the search for negative and positive influences on their sustainable performance, while in the theoretical part it clearly states psychological connections and connections with the dynamics of environmental changes. The target group of tourist guides is appropriately chosen and quantitative research was chosen for the research. Qualitative research could also be used here, which could clarify in more depth the causes and connections of the investigated influences and their impacts - it would be appropriate to mention it in the discussion or in topics for further research.

It would also be appropriate to discuss whether the examined sample of guides is representative of the presented conclusions (whether it is not just preliminary research).

Content comments:

I don't agree with "crucial" here, even if tourism is narrowed down to "tourism industry" and individual unorganized tourism is not included, an important part of the "tourism industry" are organized residential tour (typically to the sea), where guides do not apply: Being frontline employee and always playing multiple roles during the journey, tour guides group is an important and crucial part of the tourism industry. 

Formal comments:

It would be advisable to go through the entire text carefully and remove minor typos, e.g.

lack down - lock down

However, whether the crisis and uncertainty have any negative on tour guides’ psychological health and work performance still remains to be explored.

apparently missing impacts

However, whether the crisis and uncertainty have any negative impacts on tour guides’ psychological health and work performance still remains to be explored. 

Others have focused on the impact of vitality on sustainbale performance

it should probably be H2c here

H2: Vitality at work mediates the relationship between environmental dynamism and tour guides’ sustaniable performance.

the abbreviation should be entered once and theory should not be capitalized

the reference must be part of a sentence: factor that explains more than 50% of the variance, the data may exist a serious common method bias. [79]

According to Conservation of Resources (COR) Theory

second occurrence: The Conservation of Resources Theory (COR)

it would be appropriate to add a reference to Figure 1. Theoretical model in the text

Round 2

Reviewer 2 Report

Dear Authors,

I am satisfied with the improved version of your manuscript.

One remark is to correct the reference no. 57. There is a mistake in the author's name - it should be "Park, D. (...)".

My congratulations and wish you all the best.